# Central EEG Beta/Alpha Ratio Predicts the Population-Wide Efficiency of Advertisements

**DOI:** 10.3390/brainsci13010057

**Published:** 2022-12-28

**Authors:** Andrew Kislov, Alexei Gorin, Nikita Konstantinovsky, Valery Klyuchnikov, Boris Bazanov, Vasily Klucharev

**Affiliations:** 1International Laboratory of Social Neurobiology, Institute of Cognitive Neuroscience, International Laboratory of Social Neuroscience, National Research University Higher School of Economics, 101000 Moscow, Russia; 2Institute of Human Genetics, School of Medicine, Technical University of Munich, 80333 Munich, Germany; 3Operation Department, Delivery Club, 125167 Moscow, Russia

**Keywords:** neuroforecasting, neuromarketing, consumer neuroscience, EEG, eye tracking

## Abstract

Recent studies have demonstrated that the brain activity of a group of people can be used to forecast choices at the population level. In this study, we attempted to neuroforecast aggregate consumer behavior of Internet users. During our electroencephalography (EEG) and eye-tracking study, participants were exposed to 10 banners that were also used in the real digital marketing campaign. In the separate online study, we additionally collected self-reported preferences for the same banners. We explored the relationship between the EEG, eye-tracking, and behavioral indexes obtained in our studies and the banners’ aggregate efficiency provided by the large food retailer based on the decisions of 291,301 Internet users. An EEG-based engagement index (central beta/alpha ratio) significantly correlated with the aggregate efficiency of banners. Furthermore, our multiple linear regression models showed that a combination of eye-tracking, EEG and behavioral measurements better explained the market-level efficiency of banner advertisements than each measurement alone. Overall, our results confirm that neural signals of a relatively small number of individuals can forecast aggregate behavior at the population level.

## 1. Introduction

Recent neuroimaging studies have suggested that the brain activity of a group of participants can forecast the behavior of a separate and independent group of individuals [1,2,3,4,5,6,7,8,9]. In the current study, we further tested whether behavioral and neural correlates of advertisements, which had been measured in a relatively small group of participants, can predict the real (aggregated, market-level) effect of the advertisement. 

A growing number of studies have indicated the ability to use the brain activity of a group of participants to forecast the aggregate behavior of an independent and larger group of people, an approach known as *neuroforecasting*. In neuroforcasting literature, aggregate behavior refers to economy- or population-wide sums of individual behavior. The majority of such studies used functional magnetic resonance neuroimaging (fMRI) due to its sensitivity to the activities of deep brain areas [10]. Previous fMRI studies (neuro)forecasted the popularity of songs [1], microlending success [6], the effectiveness of advertisements [9,11], point-of-sales materials efficiency [8], viral marketing success [12], article virality [13], and funding success [14]. Despite the growth in the field, the overall number and scope of neuroforecasting studies remain limited (for a review, see [10]). Moreover, the number of studies in which electroencephalography (EEG) was used is even lower. The advantages of EEG include direct measurement of brain electrophysiological signals, relatively low cost of studies, high temporal resolution, and non-invasiveness. To date, EEG neuroforecasting studies have successfully predicted aggregated preferences of TV content [4], the popularity of YouTube videos [7,15,16], commercial success of movie trailers [2], and the notability of ads [3]. Additionally, neuroforecasting studies in addition to neural data often collect participant’s behavior (for example, choices to fund projects in [14], ranking of the effectiveness of the ad campaign in [5], or liking, excitability, and familiarity ratings in [11], etc.) and include it to the neuroforecasting data analysis. However, little is known about the relative accuracy of EEG, eye-tracking, and behavioral methods in forecasting aggregated (market-level) behavior. Furthermore, no neuroimaging study has examined the neuroforecasting of advertisement efficiency in digital media campaigns. In the current study, we aimed to forecast the effectiveness of 10 real banner ads using a set of behavioral (a single likability measure), eye-tracking, and neuroimaging measurements. 

Previous EEG studies have demonstrated that EEG-based metrics can predict ad recall, purchase decisions, and attitudes toward a brand at the population level (for a review, see [15]). Behavioral measurements, such as implicit reaction time (IRT), reflect attitudes toward a brand, brand images, and ad recall [17,18,19]. Furthermore, eye-tracking data also can forecast ad recall, brand recognition, and purchase intention (for a review, see [20]). By now, many consumer neuroscience studies focused on the various forms of consumer behavior (for a review, see [21]). To date, only one study [11] used a large variety of methods, such as traditional self-reports, implicit measures, eye-tracking, biometrics, EEG, and fMRI, to predict aggregate market-level effects of ads. However, that study focused on TV campaigns, but not on digital media campaigns, which are growing annually [22], making them the largest advertising medium in industrialized countries. 

Since the early 1990s, a set of EEG-based indexes was developed to track attentional engagement [23]. Beta band frequency showed up to be a reliable marker of active visual attention, being more pronounced when participants demonstrate better performance in visual attention tasks [24]. It has been suggested that stronger beta oscillations are associated with activity of the visual system, action planning, and the attentive state of the brain in general [25]. Moreover, some studies have associated medial-frontal beta oscillations with reward processing [26,27]. Such beta oscillations have been implicated in the experience of pleasure associated with a favorite brand [27]. Some evidence has suggested that beta oscillations are modulated by the brain regions connected with reward processing, including the orbitofrontal cortex [28,29,30]. On the contrary, alpha oscillations desynchronize, when participants reorient attention toward a new stimulus [31] or watch a video with frequent scene change [32]. The engagement or arousal are often evaluated as a balance between alpha and beta rhythms in gaming tasks [33], sustained attention tasks [34], and alarm-detection tasks [35]. Thus, based on prior studies, we calculated an EEG attentional engagement index as β/α-ratio to track participants’ attention during the digital media campaigns. Previous studies suggested that frontal asymmetry of alpha oscillations reflects the valence of emotion and the direction of the motivation [36,37]. Moreover, frontal alpha band asymmetry is widely used to monitor the success of advertising campaigns [11,16]. Therefore, to monitor the valence of emotion, we calculated valence index as a difference between the alpha oscillations at the left and right frontal electrodes. 

The main objective of the study is to use EEG, eye-tracking, and behavioral measure of likability to predict aggregate behaviour of a separate independent group of Internet users. Below we report two studies, where participants were exposed to 10 banners, that were also used in the digital marketing campaign of a large food retailer. The principle results demonstrated that the β/α-ratio and total time the participant fixated at the image, that were recorded in the lab, significantly correlated with the aggregate behaviour of a separate independent group of Internet users. Overall, we concluded that a combination of EEG, eye-tracking, and behavioral measurements better explained the variation in the aggregate behaviour than each measurement alone.

## 2. Materials and Methods

To neuroforecast the aggregate behaviour, we used EEG as one of the most popular neuroscientific techniques for marketing studies [38]. EEG is a safe and noninvasive method. Many EEG systems are portable and can be combined with eye-tracking. Furthermore, EEG has the high temporal resolution that makes it suitable for further studies of dynamic marketing stimuli. In addition, we used eye tracking that allows to identify which items capture visual attention [39]. In order to collect (a) neuroscientific measures and (b) behavioral data, which cannot be collected in the one study due to some differences in study designs, two consecutive studies been performed. Thus, in Study 1 we collected EEG and eye-tracking data; in Study 2 we collected behavioral data to forecast the outcome of the real digital marketing campaign of a large food retailer.

### 2.1. Study 1: EEG and Eye-Tracking Study

#### 2.1.1. Participants

We recruited 26 participants via a panel service. All participants reported no history of neurological disorders, no history of psychotropic drugs, and no substance abuse in the past month; all had normal or corrected-to-normal vision and no colorblindness. Participants represented the target audience for the digital marketing campaign of a premium food retailer: they declared upper-middle or high-income, age range of 21–35 years, and at least one visit per week to a premium retailer (median = 3). All participants provided written informed consent and were naïve to the main purpose of the study. At the beginning of the experiment, each participant was informed about the experimental procedure, the eye-tracking, and EEG methods, and was instructed to look at the screen during the experiment without an explicit behavioral task. One participant was excluded from the data analysis because of extensive eye movements. Thus, the final data set included 25 participants (16 females, median age = 29 years). The study protocol was approved by the ethics committee of the local university. 

#### 2.1.2. Stimuli

To make our study more ecologically valid, we used the original stimuli used in the marketing campaign. Importantly, all banners had the same structure (a similar organization of the elements and the same size of the elements), that made them comparable to each other. The stimuli consisted of 10 colored banner ads designed by a premium food retailer for a digital media campaign. The stimuli were presented on a 17” LED monitor (screen resolution = 1920–1080 pixels) using the SMI Experiment Centre (SensoMotoric Instruments GmbH, Teltow, Germany). At the beginning of each trial, a banner ad was presented for 5 s (Figure 1A). Next, the participant answered a short filler question using a 5-point Likert scale. Filler questions probed attitudes toward the brand and banner ad; filler questions were not analyzed in the current study. 

During Study 1 (A), participants observed a banner ad (5 s) and answered a filler question. At the end of the trial, the participants rated the posters. During Study 2 (B), participants observed a banner ad (5 s) and statement (e.g., “I like it”) (500 ms). At the end of the trial, participants indicated whether they agreed or disagreed with the statement. In both studies, the intertrial interval (ITI) was 2–6 s. 

During each trial, EEG and eye-tracking data were recorded simultaneously. Each trial lasted approximately 30 s, including a variable intertrial interval (2–6 s). Overall, 30 trials were presented in a random order during each study with 3 repetitions per banner. Each study lasted approximately 15 min. Each banner ad consisted of pictorial, textual, and brand elements; elements were similarly spatially arranged in all banner ads (Figure 3). Importantly, all banners differed in their pictorial and textual elements. 

#### 2.1.3. Aggregated Market-Level Effects of Ads

In the current study, we aimed to neuroforecast the market-level outcome of the real digital marketing media campaign of the large food retailer. This media campaign aimed to motivate consumers to visit the retailer’s webpage and spend longer time. Therefore, the effect of the media campaign was measured as time (mean session duration) that Internet users spent at the retailer’s webpage after clicking the banner ad. For two months, 10 banner ads were randomly aired on various websites (4 h each day) using a programmatic platform Hybrid (https://hybrid.ai/, accessed on 20 February 2020): each banner was randomly presented at the different websites with equal probability to the preselected pool of Internet users with higher-than-average income. Online statistics indicated that 291,301 Internet users (age: 20–45) were exposed to banner ads 1,456,182 times. Overall, during the digital campaign, each Internet user was exposed approximately 5 times to the banner ads, which had been chosen randomly for each exposure. To compare the relative effectiveness of each banner ad, we calculated a session duration index (SD-index) that indicated how much time on average users spent on a retailer’s webpage after clicking a certain banner ad, which is a proxy of the market-level interest, which this banner has evoked. Figure 2 illustrates the distribution of the SD index across 10 banner ads. 

#### 2.1.4. Eye-Tracking Recordings and Analysis

To study whether the attention to the banners can predict aggregate efficiency of banners, we used the eye-tracking technique. Eye-tracking data were recorded using SMI RED-m (SensoMotoric Instruments GmbH, Teltow, Germany) with a sampling frequency of 60 Hz and an accuracy level of 0.5 degrees. The participants were seated at a table in front of a computer screen, and their seating position was adjusted to ensure that they remained centered in front of the monitor at a distance less than 65 cm. Standard methods (velocity- and acceleration-based) were used to segment the gaze trajectories into a sequence of fixations and saccades. All fixations outside the range of 50–600 ms were removed as outliers (less than 5%). 

Each stimulus/banner ad depicted three areas of interest (AOIs): pictorial, textual, and brand elements (Figure 3). To compare the eye-movement characteristics of participants for each banner ad, we calculated the dwell time index (DT index) for each AOI that indicated the total time the participant fixated at the specific element (for a similar approach, see [40]). Thus, for each banner, the following three DT indexes were calculated: DT picture index, DT text index, and DT brand index. Eye-tracking data were analyzed using SMI BeGaze 3.7.59 software (SensoMotoric Instruments GmbH, Teltow, Germany). 

#### 2.1.5. EEG Recording

To study whether an EEG-based engagement index can predict the aggregate efficiency of banners we recorded the EEG from 28 scalp electrodes. The EEG data were collected using the NVX36 amplifier (Medical Computer System) with a 500 Hz sampling rate. The following 28 Ag/AgCl electrodes were positioned according to the international 10–20 system: Fp1, Fp2, F3, F4, C3, C4, P3, P4, O1, O2, F7, F8, T7, T8, P7, P8, Fz, Cz, Pz, Oz, FC1, FC2, CP1, CP2, FC5, FC6, CP5, and CP6. EEG signals were referenced to arithmetically linked mastoids. Impedance was kept below 5 kΩ. The electrooculogram was recorded with electrodes placed at the left outer canthi and below the right eye. To synchronize EEG and eye-tracking data EEG amplifier collected event markers from the stimulation software (SMI Experiment Centre, SensoMotoric Instruments GmbH, Teltow, Germany).

#### 2.1.6. EEG Analysis 

We performed EEG preprocessing using the Brainstorm toolbox [41]. First, the raw data were visually inspected for artifacts, and the noisy segments were removed. Second, the EEG data were filtered with a 1 Hz high-pass filter (slope = 48 dB/oc) and a 40 Hz low-pass filter. Third, to correct for eye-movement artifacts, we used JADE independent components analysis (ICA): the eye-movement artifacts were removed according to their topography and correlation with the EOG, as implemented in the Brainstorm toolbox. Fourth, the data was re-referenced against the average activity at all electrodes and then filtered in alpha (8–12 Hz) and beta (16–24 Hz) bands. Fifth, the continuous data were segmented for each banner ad into 10 segments (−500 ms to 5000 ms). Lastly, the valence index and engagement index for each segment were calculated, as described below. To analyze alpha (8–12 Hz) and beta (16–24 Hz) oscillations, we applied fast Fourier transform (FFT) over 4000 ms of post stimulus intervals, acquiring mean power across time in alpha (8–12 Hz) and beta (16–24 Hz) bands. Therefore, in the current study, for each banner, we calculated a frontal asymmetry index—valence index (F4(α) minus F3(α))—as a difference between the power levels of the alpha rhythm at the F4 and F3 electrodes and the engagement index (Beta (Cz + Pz + P3 + P4)) / (Alpha (Cz + Pz + P3 + P4) as a sum of beta/alpha ratio at the central electrodes. The resulting data has been averaged across trials (type of the banners), and to obtain the cumulative index, across the subjects.

### 2.2. Study 2: Behavioral Study

To collect additional behavioral data, we conducted Study 2. The experimental protocol was largely identical to Study 1, except for minor modifications to the trial structure (Figure 1B).

#### 2.2.1. Participants

We recruited 48 participants (25 females, median age = 31) via a panel service. Participants had socioeconomic characteristics similar to those recruited in Study 1.

#### 2.2.2. Study Design

In behavioral Study 2, we used the same 10 banner ads as in Study 1. Figure 1 shows that each trial started with a 5000 ms presentation of a banner followed by the statement, “I like this ad” in order to measure ‘likeability’ since its wide use in practice. Next, after the statement, two response options (“yes” and “no”) appeared on the screen, with a delay of 500 ms (to avoid responses of participants, who intend to just to click through and finish as fast as possible). At the end of the trial, the participants indicated their choice using the keyboard. A total of 20 trials (10 banner ads × 2 repetitions) were randomly presented for approximately 5 min. The study was conducted online using FasTest (FasTest Neuro Solutions, Inc., version 2021, Lewes, DE, USA) software. 

#### 2.2.3. Behavioral Data Analysis

First, all reaction times longer than 2500 ms were removed as outliers [42]. 

Furthermore, the likeability index was calculated as the proportion of participants who agreed with the statement (“I like this ad”).

#### 2.2.4. Multiple Regression Models

Using multiple linear regression models, we further examined whether behavioral (Study 2), eye-tracking and/or EEG data (Study 1) could predict behavior at the population level. We used the aggregate SD index of the banners’ efficiency (for 291,301 Internet users) as the dependent variable and up to four independent variables. The following three models were tested to explore the various predictors of the *SD index* that indicated how much time on average users spent on a retailer’s webpage after clicking a certain banner ad: (1)Model I (null model) included only the behavioral likeability index as a predictor.(2)Model II included the EEG-based valence index and engagement index as predictors.(3)Model III included eye-tracking-based DT picture index, DT text index, and DT brand index as predictors.(4)Model IV included all psychophysiological independent variables (valence index, engagement index, DT-picture index, DT-text index, DT-brand index) as predictors.

Similar to the seminal neuroforecasting study [6], we also used Pearson’s correlation analysis to study the relationship of the engagement index (β/α ratio), valence index (α asymmetry), DT-indexes (dwell time for picture, text, or brand elements of banners), and likeability index with the population-level session duration index in more detail. The results are presented in Appendix A. Visual summary of the investigated variables, together with their connection to the AIDA framework, widely used in advertising, is presented in Figure A1.

## 3. Results

Separate linear regression models estimated whether (a) a behavioral likeability index, (b) an EEG-based valence index (α asymmetry), engagement index (β/α ratio), and (c) eye-tracking-based DT indexes (DT picture index, DT text index, and DT brand index) were predictive of behavior at the population level (see Table 1 and Figure 4). The adjusted R^2^, the coefficient of determination, and Akaike Information Criterion (AIC) indicated the better fit of Model IV that combined all the psychophysiological independent variables (adj R^2^ = 0.79, p = 0.03, AIC = 86) compared to Model I (adj R^2^ = 0.09, p = 0.2, AIC = 99), Model II (adj R^2^ = 0.45, p = 0.04, AIC = 95), and Model III (adj R^2^ = −0.05, p = 0.5, AIC = 102), despite penalties for additional predictors. Overall, a combination of EEG and eye-tacking indexes (Model IV) explained 79% of the variance in aggregate population behavior indicated by the SD-index, whereas the EEG-based engagement index alone explained only 43% of the variance in the population-level behavior.

## 4. Discussion

Our findings further support the view that the neural activity of a limited number of people can predict decisions at the population level. In Study 1, we analyzed the EEG activity and eye movements of 26 participants while they observed 10 banner ads from the real digital marketing campaigns of a large food retailer. In a separate behavioral study, Study 2, another 48 participants observed the same ads and took a behavioral test. As a principal result we can report that linear regression modeling indicated that the combination of the EEG-based valence index (α asymmetry), engagement index (β/α ratio), and eye-tracking-based DT indexes (DT picture index, DT text index, and DT brand index) could explain the variance of the aggregated (population level) effect of banners better than each predictor separately. 

Advertising might influence consumers’ emotions, memory or behavior. The effects of advertising can be hierarchical [43,44]: the lower-order effects could be necessary preconditions for the later higher-order effects. Therefore, measuring the effectiveness of an advertising campaign is a challenge for practitioners and scientists [45]. Capturing customers’ attention is critical to marketing success [46,47,48,49], particularly in the visually overloaded online environment. In the current study, the effect of the media campaign was measured as time (mean session duration) that Internet users spent at the retailer’s webpage after clicking the banner ad. Importantly, a marketing campaign, that was used in our study, aimed to increase time spend during website visit, after clicking the banner. As global media spends rise over the years [22], marketers are calling for accurate assessments of advertising effectiveness [39], because traditional non-physiological measures of advertising effectiveness have strong limitations. For example, consumers often state their preferences incorrectly [50]. Thus, in recent decades, consumer neuroscience has developed not only as an academic field but also as a marketing research practice. In particular, neuroforecasting studies have demonstrated that the results of neuroimaging lab studies can be translated into real-life aggregated behavior. Thus, neuroimaging may provide marketeers with additional information that supplements conventional marketing research methods [39,51].

In the current study, statistically significant regression models used either EEG-based indexes (adj *r*^2^ = 0.45) or EEG- and eye-tracking-based indexes (adj *r*^2^ = 0.79) as independent variables. However, regression models that used only eye-tracking data or behavioral measures were not significant. The coefficient of multiple determination for multiple regression (r^2^) in the current study was consistent with previous EEG-based neuroforecasting studies [2,7,16,52], in which the percentage of the dependent variable variation that a linear model explained ranged from 30% to 70% (for review, see [15]). Overall, our findings support the idea that neurophysiological measures can better account for aggregated behavior than traditional behavioral measures [5,6,9,10,11,14,15]. 

Our multiple regression modeling (Model IV) showed that the engagement index (EEG-based frontal beta/alpha ratio) and DT-picture index (total time the participant fixated at the picture AOI) were particularly predictive for the aggregated effect of advertising. This is in line with previous studies that showed a connection between measures of engagement or arousal and consumers’ responses to advertising [53], ad notability [3], and click-through rate [52] at the population level. Interestingly, EEG beta activity has been previously implicated in positive brand experience [27], reward processing [26,54], and population-wide movie preference [2]. 

In the current study, the valence index, measured as frontal EEG alpha asymmetry, was not significantly predictive of the population-level effect of advertising. The affect–integration–motivation (AIM) framework suggests that the “affective” responses, which underlie consumers’ choices, might broadly generalize across individuals, and thereby may reveal “hidden information” regarding mass preferences and aggregated choices (40, 50). Many previous neuroforecasting fMRI studies have focused on the ventral striatum, whose role in reward processing is strongly dependent on the saliency associated with reward [55,56]. Thus, we can speculate that the EEG-based engagement index—frontal beta/alpha ratio—may depict the saliency associated with the product/banner that could correlate with aggregated market-level preferences. 

We also showed that eye-tracking-based DTI-picture index (with product picture), but not DTI-text and DTI-brand (logo) indexes, significantly predicted the aggregated effect of advertising. Similarly, in a previous study [57], gaze fixations on product elements (i.e., phone) were positively correlated, whereas gaze fixations on brand elements (i.e., logo) showed a negative correlation. Furthermore, a positive connection between fixations on the packaging design elements and consumers’ choices has been previously reported [58].

Our study also makes practical implications for the field of consumer research. We showed that eye-tracking in combination with EEG is particularly effective in predicting the effects of advertisements. We also confirmed that a combination of eye-tracking and EEG better explained the market-level effects of advertisements than each measurement alone.

The current study has several important limitations; most of them were determined by the neuroforcasting study design, and by commercial information disclosure limitations. For example, we had a limited access to the banner’s efficiency data. The restricted timing of psychophysiological data acquisition also limited the number of stimulus repetitions during Study 1. We invited participants who matched an average customer profile. Thus, the generalization of our results to the total population must be approached with caution. We also presented advertising banners in lab settings outside the native context of digital media. Therefore, the ecological validity of our findings should be further verified using other study designs. Notably, due to the limitation of the real digital marketing campaign, we used only 10 banners provided by the food retailer. Thus, our statistical analysis was limited by the relatively small dataset. Importantly, in real practice, marketers rarely run digital marketing campaigns with more than 10 different banner ads. Therefore, real-life practice often limits the datasets of aggregated behavior available for neuroforecasting studies. Nevertheless, we believe this study may extend our knowledge in the field of neuroforecasting.

To conclude, we fulfilled the main objective of the study and confirmed that EEG, eye tracking, and behavioral measure of likability can predict aggregate behaviour of a separate independent group of Internet users. The multiple linear regression models demonstrated a significant association of psychophysiological measures (especially the beta/alpha ratio) with ad efficiency at the population level. Importantly, a combination of EEG and eye-tracking data better explained ad efficiency than the behavioral likability measure. Thus, our results provide an additional step in verifying the neuroforecasting approach in the context of online digital media campaigns.

## Figures and Tables

**Figure 1 brainsci-13-00057-f001:**
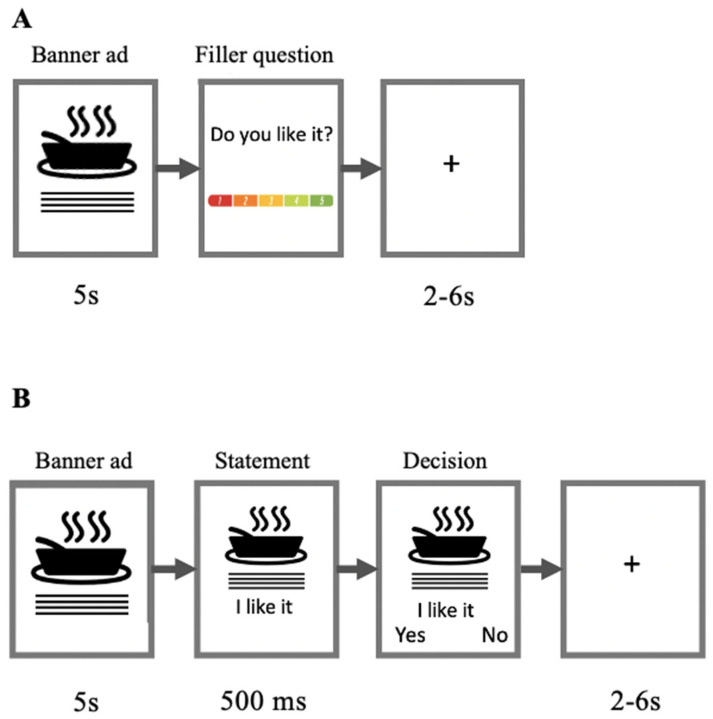
Experiment design: Trial structure of Study 1 (**A**) and Study 2 (**B**). During Study 1 (**A**), participants observed a banner ad (5 s) and answered a filler question. At the end of the trial, the participants rated the banners. During Study 2 (**B**), participants observed a banner ad (5 s) and statement (e.g., “I like it”) (500 ms). At the end of the trial, participants indicated whether they agreed or disagreed with the statement. In both studies, the intertrial interval (ITI) was 2–6 s.

**Figure 2 brainsci-13-00057-f002:**
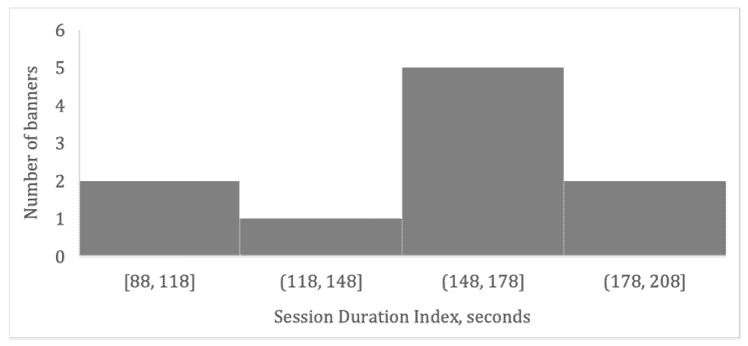
Population-level effects of ads. Distribution of the session duration index (SD index) across 10 banner ads.

**Figure 3 brainsci-13-00057-f003:**
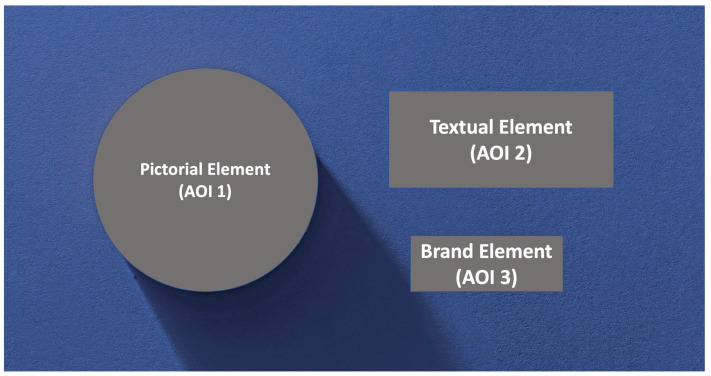
Each banner ad consisted of three key elements: pictorial, textual, and brand elements. Three areas of interest (AOI) were selected for eye-tracking analysis: pictorial element (AOI-1) consisting of the picture of the product, textual element (AOI-2) consisting of the endorsing text, and brand element (AOI-3) containing the logo of the brand. The size AOI-3 was exactly the same across all banners, whereas the sizes of AOI-1 and AOI-2 differed across banners by less than 5%.

**Figure 4 brainsci-13-00057-f004:**
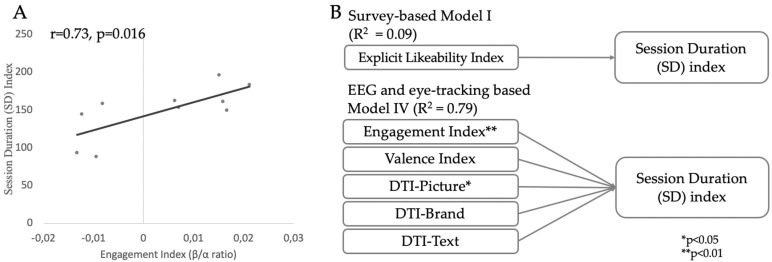
Relationship between population-level effects of banner ads—session duration (SD) index and behavioral likability index, EEG, eye-tracking data. (**A**) The scatterplot indicates the session duration (SD) index as a function of the averaged engagement index (β/α ratio). The line represents a linear trend estimate. (**B**) The diagrams show key results of linear regression models that estimated whether survey-based explicit likeability index, EEG-based valence and engagement indexes, eye-tracking-based DT picture, DT text, and DT brand indexes were predictive of the session duration index. The adjusted R^2^ value is given in parentheses; asterisks indicate significant coefficients (* *p* < 0.05, ** *p* < 0.01).

**Table 1 brainsci-13-00057-t001:** Results of the linear regression models predicting population effects of banner ads—session duration index—using behavioral, electroencephalography (EEG), and eye-tracking data (Dwell Time Indexes, DTI). Confidence interval values are presented in the square brackets.

	Model I	Model II	Model III	Model IV
Likeability index (behavioral)	15 (−10 40)			
Engagement index (EEG)		25 * (5 45)		30 ** (13 47)
Valence index (EEG)		−6 (−26 13)		−3.5 (−20 13)
DT picture index (eye-tracking)			17 (−38 72)	34 * (4 65)
DT text index (eye-tracking)			−2 (−52 48)	20 (−10 51)
DT brand index (eye-tracking)			1 (−33 36)	3 (−15 21)
Adjusted *R*^2^	0.09	0.45	−0.05	0.79
AIC	99	95	102	86
*p*-value	0.2	0.04	0.5	0.03

* indicates *p* < 0.05, ** indicates *p* < 0.01.

## Data Availability

Data supporting reported results can be found at https://docs.google.com/spreadsheets/d/1wSjsmgMQZtn_ajUynRD1CSD2sDTzX5Vm/edit#gid=861306597 (accessed on 22 December 2022).

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
