# Peer review of "Central EEG Beta/Alpha Ratio Predicts the Population-Wide Efficiency of Advertisements"

_brainsci, 2022, doi:10.3390/brainsci13010057_

Round 1

Reviewer 1 Report

It was very interesting to read your contribution "Central EEG beta/alpha ratio predicts the population-wide efficiency of advertisements".

I suggest making some minor changes and/or additions

-Figure 3 contains an error in the nomenclature. AOI3 (Brand element) is renamed in the figure incorrectly as AOI2 (brand element)

-If possible, to facilitate understanding, add a table listing and summarizing the demographic and sociocultural characteristics of the sample examined in study 1 and study 2

-Add the specification of how the EEG and Eye tracker data were synchronized

Author Response

Point 1: It was very interesting to read your contribution "Central EEG beta/alpha ratio predicts the population-wide efficiency of advertisements".

I suggest making some minor changes and/or additions

-Figure 3 contains an error in the nomenclature. AOI3 (Brand element) is renamed in the figure incorrectly as AOI2 (brand element)

Response 1: We are sorry for the confusing error. We have changed the picture accordingly.

Point 2: If possible, to facilitate understanding, add a table listing and summarizing the demographic and sociocultural characteristics of the sample examined in study 1 and study 2.

Response 2: Thank you for the important question. For the current study, we selected participants, who  matched retailer’s target audience as defined by the retailer. The inclusion criteria included age, income and number of visits to a premium retailer per week. Therefore, we invited 21–35 years old participants with a upper-middle or high-income, who visits a premium retailer at least one visit per week. Unfortunately, due to the strict local ethical guidelines for collaborative research with private companies we were not able to collect other demographic and sociocultural characteristics of the participants.

Point 3: Add the specification of how the EEG and Eye tracker data were synchronized

Response 3: : Indeed, combination of EEG and eye tracking is technically challenging. Therefore, we set up shared event markers that appeared in both the EEG and the eye tracking data at the same time. We marked events in the eye tracking software (SMI Experiment Centre software, SensoMotoric Instruments GmbH) and sent pulses to mark stimulus events in EEG recorder. We added this information to lines 213-215.

Reviewer 2 Report

First, I want to congratulate the authors for their excellent work. My comments on the paper will serve to improve mainly the theoretical argumentation component and the research model.

Introduction:

Missing: 1) The main objectives of the study; 2) Principal results; 3) and, Main conclusions.

 Literature review:

What is meant by advertising effectiveness?

What do behavioral methods mean in the prediction of aggregate behavior?

What typologies of behavioral methods do you refer to?

What is meant by aggregate behavior?

The statement in lines 56 and 57 should be reworded because there are more studies that use a wide range of neuroscience methods applied to consumer reactions, I advise the authors to take a look at the following papers:

Cardoso, L.; Chen, M.-M.; Araújo, A.; de Almeida, G.G.F.; Dias, F.; Moutinho, L. Accessing Neuromarketing Scientific Performance: Research Gaps and Emerging Topics. Behav. Sci. 2022, 12, 55. https://doi.org/10.3390/bs 12020055 Academic Editor: Joseph

Sousa, C.; Lara, E.J. State of the art of national and international publications on neuromarketing and neuroeconomics. Braz. J. Mark.—BJM Rev. Bras. Mark. 2016, 15, 28–41.

 Methodology

 The research method is absent in the paper. As it seems to me that it is the quasi-experimental method, I find it convenient to justify it. They should also justify the reasons why they are using this method, clearly justify the decisions made for the choice of stimuli used (scientifically justified), clearly justify why they used and what they want to show with the technologies applied (i.e. align to the results obtained) and, most importantly, how they intend to show the effectiveness of advertising, and by means of which variables and theoretical models.  It would enrich your work and even make it more straightforward if you could present a visual model (a figure, for example) to justify what I have previously mentioned (It is not clear what scientific models you use for behavior, engagement, valence, you do not explain the origin of the DT-index, etc.)

It is also unclear the different phases of the study and why.

Discussions, implications, and conclusions

This department is excellent and theoretically supported however, the conclusions are absent, and conclusions should relate back to the introduction section, the hypothesis.

Also absent the implications, the significance of your results or practical application.

The limitations are clear.

Author Response

Point 1 First, I want to congratulate the authors for their excellent work. My comments on the paper will serve to improve mainly the theoretical argumentation component and the research model.

Introduction:

Missing: 1) The main objectives of the study; 2) Principal results; 3) and, Main conclusions.

Response 1: As requested, we added the main objectives (lines 94-100), principal results (lines 297); and main conclusions (lines 379-384).

Point 2:  Literature review: What is meant by advertising effectiveness?

Response 2: Advertising might influence consumers' cognitions, emotions, memory or behavior. The effects of advertising can also be hierarchical (Lavidge and Steiner, 1961; Vakratsas and Ambler 1999): the lower-order effects could be necessary preconditions for the later higher-order effects. Capturing customers’ attention is critical to marketing success (Davenport and Beck, 2001), particularly in the visually overloaded online environment. In the current study the effect of the media campaign was measured as time (mean session duration) that Internet users spent at the retailer’s webpage after clicking the banner ad. A marketing campaign, that was used in our study to calculate aggregate behavior, aimed to increase time spend during website visit, after clicking the banner. We did not measure other possible effects of ads because no additional information was available from our industrial partner. We added relevant details to the manuscript, lines 165-166 and 301-307.

Point 3: What do behavioral methods mean in the prediction of aggregate behavior?

Response 3: : We are sorry for the missing details. We used a behavioral likeability index - a proportion of participants who agreed with the statement (“I like this ad”). Neuroforecasting studies often in addition to neural data also collect participant’s behavior (for example, choice to fund projects in Genevsky et al (2017), ranking of the effectiveness of the ad campaign in Falk et al (2012), liking, excitability, and familiarity ratings in Venkatraman et al (2012), etc). Neurofocasting approach suggests that individual choices may, in addition to individual brain data, provides partial information about future market behavior (Knutson and Genevsky, 2018). Therefore, our multiple linear regression models included the behavioral likeability index. In the new version of the manuscript, we have changed the Introduction (lines 52-55) and the rest of the text whenever it was needed.

Point 4:  What typologies of behavioral methods do you refer to?

Response 4: As we mentioned above, neuroforecasting studies usually collect participant’s behavior (for example, choices to fund projects in Genevsky et al (2017), ranking of the effectiveness of the ad campaign in Falk et al (2012), or liking, excitability, and familiarity ratings in Venkatraman et al (2012), etc). In the current study we used a behavioral likeability index - a proportion of participants who agreed with the statement (“I like this ad”). For the detailed changes made in the new version of the manuscript please see our answer to your previous question. 

Point 5: What is meant by aggregate behavior?

Response 5: : In neuroforcasting literature, aggregate behavior refers to economy- or population-wide sums of individual behavior. We added the definition of the aggregate behavior to the new version of the manuscript, see lines 42-43.

Point 6: The statement in lines 56 and 57 should be reworded because there are more studies that use a wide range of neuroscience methods applied to consumer reactions, I advise the authors to take a look at the following papers:

Cardoso, L.; Chen, M.-M.; Araújo, A.; de Almeida, G.G.F.; Dias, F.; Moutinho, L. Accessing Neuromarketing Scientific Performance: Research Gaps and Emerging Topics. Behav. Sci. 2022, 12, 55. https://doi.org/10.3390/bs 12020055 Academic Editor: Joseph

Sousa, C.; Lara, E.J. State of the art of national and international publications on neuromarketing and neuroeconomics. Braz. J. Mark.—BJM Rev. Bras. Mark. 2016, 15, 28–41.

Response 6: We, fully agree, that there are much more studies in consumer neuroscience, which been applied to consumer reactions. Therefore, we added to lines 62-63 new references to reviews in consumer neuroscience. Importantly, in the current study we particularly focus on the studies predicting consumer behavior at the population level.

Point 7:  Methodology: The research method is absent in the paper. As it seems to me that it is the quasi-experimental method, I find it convenient to justify it. They should also justify the reasons why they are using this method, clearly justify the decisions made for the choice of stimuli used (scientifically justified), clearly justify why they used and what they want to show with the technologies applied (i.e. align to the results obtained) and, most importantly, how they intend to show the effectiveness of advertising, and by means of which variables and theoretical models.  It would enrich your work and even make it more straightforward if you could present a visual model (a figure, for example) to justify what I have previously mentioned (It is not clear what scientific models you use for behavior, engagement, valence, you do not explain the origin of the DT-index, etc.)

Point 7.1: «justify the reasons why they are using this method»

Response 7.1: We used EEG as one of the most popular affordable neuroscientific techniques for marketing studies that has many advantages (Bazzani et al., 2020). It is a safe and noninvasive method.  Many EEG systems are portable and can be combined with eye-tracking. Furthermore, EEG has the high temporal resolution that makes it suitable for studies of dynamic marketing stimuli. Eye tracking is also a popular research tool in marketing studies that allows to identify which items capture someone's attention (Wedel, 2015). Also, there is relatively little number of neuroforecasting studies, which combine both on the eye-tracking and EEG measures. As requested, we added additional motivation, see lines 104-108.

Point 7.2: «justify the decisions made for the choice of stimuli used (scientifically justified)”

Response 7.2: To make our study more ecologically valid, we used the original stimuli used in the marketing campaign. Importantly, all banners had the same structure (a similar organization of the elements and the same size of the elements), that makes them comparable to each other.  As requested, we added additional motivation, see lines 144-146.

Point 7.3: « clearly justify why they used and what they want to show with the technologies applied (i.e. align to the results obtained) and, most importantly, how they intend to show the effectiveness of advertising, and by means of which variables and theoretical models.

Response 7.3: We apologize for the missing details in the previous version of the manuscript. To address your comment, we added a theoretical model, linking the AIDA model with our variables and population-wide ads efficiency indicators. The AIDA model refers to Attention, Interest, Desire and Action, respectively to highlight the set of cognitive process that may occur during an exposure to advertisement. According to the AIDA model, the aim of marketing is to attract the attention from consumers, stimulate their interest and desire to the final buying action. In the AIDA model framework, eye-tracking data may correspond to “Attention” stage of the model. EEG-based engagement index might correspond to both “Attention” and “Interest” stages of the model; while EEG-based valence index might correspond to both “Interest” and “Desire” stage, while behavioral likeability data might correspond to “Desire” stage of the AIDA model. Finally, a session duration index (SD-index) that indicated how much time on average users spent on a retailer’s webpage after clicking a certain banner ad, and retailer sales (not measured in the current study) may resemble the “Desire” and “Action” stage, correspondingly. Altogether, variables in our study were able to forecast aggregate consumers’ actions.

Please note that this visual model was made only for illustrative purposes.

We added this model to Appendix B.

Point 7.4: It is also unclear the different phases of the study and why.

Response 7.4: To answer your comment, we added missing information, see lines 109-112.

Point 8: This department is excellent and theoretically supported however, the conclusions are absent, and conclusions should relate back to the introduction section, the hypothesis.

Response 8: We agree with this comment. We added the conclusion that relates back to the introduction section. See lines 381-386.

Point 9: Also absent the implications, the significance of your results or practical application.

Response 9: Our study also makes practical implications for the field of consumer research. We showed that eye-tracking in combination with EEG is particularly effective in predicting the market-level effects of advertisements. We also confirmed that a combination of eye-tracking, EEG and behavioral measures better explained the market-level effects of advertisements than each measurement alone. We added the text to lines 349-352.
